



# The impact of microplastic weathering on interactions with the soil environment: a review.

Frederick Büks[1], Martin Kaupenjohann[1]

[1]Chair of Soil Science, Dept. of Ecology, Technische Universität Berlin, 10587 Berlin, Germany

*Correspondence to:* Frederick Büks (frederick.bueks@tu-berlin.de)

**Abstract.** Recent studies have reported the influence of microplastic on soil quality parameters. Mass concentrations of plastic particles as found in highly contaminated soils were shown to weaken the soil structure by reducing the proportion of water stable aggregates (WSA). In addition, parts of the edaphon are adversely affected by mainly the <100 µm microplastic fraction. The specific interaction of soil microplastic with other

particulate organic matter (POM) and the mineral phase during the formation of soil aggregates as well as the adverse effects of especially the small-sized fraction, which has low weight but high specific surface area, justify a focus on surface properties of the soil microplastic and their alteration during the plastic life cycle. Exposed to UV radiation, juvenile plastic undergoes photochemical weathering with embrittlement and the formation of surface

charge. When plastic particles enter the soil environment, a second step takes place, that includes biogeochemical weathering with enzymes, biotic and abiotic acids, oxidants as well as bioturbation and feeding of the soil fauna. This work integrates recent findings on the effects of microplastic on soil structure and biota, the genesis of its surface characteristics and discusses how to reproduce them to conduct laboratory experiments with close-to-nature

designer microplastic.



## 1 Our legacy of microplastic

The mass production of plastic articles of daily use started in the early 1950[th] (Thompson et al., 2009). Until today, a broad variety of plastics and derivatives has entered the markets leading to an all-time industrial output of 8300 Mt and an annual production of 380 Mt in 2015 as well as an alarming release into the environment (Geyer et al., 2017). Widespread studies could show that today ecosystems such as inland and coastal waters, sediments, the open and deep seas, soils and even the atmosphere of remote areas are contaminated with microscopic plastic fragments (Cole et al., 2011; Woodall et al., 2014; Wu et al., 2018; Büks and Kaupenjohann, 2020; Trainic et al., 2020).

When plastic resources are dumped or dissipated into the terrestrial environment, recycling becomes difficult leading to accumulation, since the material is comminuted but hardly degraded. Only roughly estimated is the today's amount of microplastic introduced into soils. Inputs occur through specific entry pathways like littering and dispersion from landfills, the application of wastewater, contaminated surface water, sewage sludge, composts, digestates, mulching foils and coated fertilizers, road dust as well as atmospheric deposition (Eerkes-Medrano et al., 2015; Huerta Lwanga et al., 2017a; Weithmann et al., 2018; Corradini et al., 2019; Dierkes et al., 2019; He et al., 2019; Edo et al., 2020; Huang et al., 2020; Katsumi et al., 2021; Szewc et al., 2021). Estimations of the amount of microplastic brought into soils by the agricultural application of sewage sludge range from 0.3 to 20 mg kg$^{-1}$ dry soil (Nizzetto et al., 2016; Büks et al., 2020b). Field campaigns found a predominance of small-sized microplastic <250 µm, common average concentrations of about 1 mg kg$^{-1}$ dry soil and values multiple orders of magnitude above in highly contaminated areas (Büks and Kaupenjohann, 2020). Material composition of plastic residues is strongly determined by locality, adjacent land use as well as the set of contamination pathways and appears to comprise mainly the most produced plastic types polyethylene (PE), polypropylene (PP), polyvinyl chloride (PVC), polyethylene terephthalate (PET), polyurethane (PU) and polystyrene (PS) (Büks and Kaupenjohann, 2020).

Recent studies pointed out, that the microplastic introduced into soils has the potential to influence soil physiochemical and biological characteristics. However, the physiochemistry of soil is largely a physiochemistry of surfaces, and most of the underlying laboratory experiments used unweathered microplastic. These particles with juvenile surface characteristics are not supposed to be fully representative for plastic found in the environment, which underwent photooxidative and also complex biogeochemical alteration and, thus, interact differently with soil organic matter (SOM), the mineral phase and soil biota. To the best of our knowledge, there is no study using a pre-treatment of experimental soil microplastic to strictly imitate this natural weathering pathway. The aim of this work is to collect data on the effect of microplastic surface characteristics on soil structure and soil life and integrate our knowledge about the photooxidative and biogeochemical phase of



weathering in order to better reproduce surface characteristics of environmental microplastic in future laboratory and field experiments.

## 2 Search pattern

The Web of Science Core Collection database was searched for studies focusing upon the
effect of microplastic on soil structure published until July 2021. A pattern of search terms was established, combining common terms related to soil structure (aggregate stability, aggregate structure, macroaggregate*, microaggregate*, water stable aggregates OR WSA, water holding capacity OR WHC, saturated hydraulic conductivity OR SHC, bulk density, compactibility and penetration resistance) with plastic type (plastic, microplastic, nanoplastic
as well as the most produced plastics polyethylene OR PE, polyethylene terephthalate OR PET, polypropylene OR PP, polystyrene OR PS, polyvinyl chloride OR PVC, polyurethane OR PU and the common textile materials polyamide OR PA, polyacrylic acid OR PAA and polyester OR PES). The pattern was applied to the database taking into account title, abstract text and a restriction to entries containing the word "soil". Studies not related to plastic
pollution in soils, studies on biodegradation of intact plastic mulch foils as well as studies with use of only macroscopic objects >5 mm were excluded. Further plastic types occurring within the studies were also included into the review. The effect of microplastic on the soil fauna as well as the present diversity of different methods to weather microplastic surfaces in soil biological studies were discussed based on data collected in a recent comprehensive review
(Büks et al., 2020a).

## 3 Interference with soil structure

Studies of the past few years demonstrated, that high concentrations of microplastic particles could alter soil structural characteristics by influencing aggregate formation dynamics
(Table 1). Shape, size and type of microplastic as well as soil environmental conditions are thereby found to be variables of this effect. De Souza Machado et al. (2018) demonstrated, that by application of 0.5-20 g of juvenile plastic per kg dry soil larger fragments (160-1200 µm) appeared to be bound only loosely within re-aggregated soil samples, whereas microbeads (15-20 µm) and fibers were more integrated into rebuilding macroaggregate
structure. This different occlusion dynamics has come along with a not consistently clear pattern of reduced bulk density, increased water holding capacity (WHC) due to decompaction as well as fewer water stable aggregates (WSA). The reduction of WSA is confirmed by some studies incubating fibers and microbeads within a similar particle size range for up to 70 days (de Souza Machado et al., 2019; Liang et al., 2019) and a
comprehensive examination of soil structure compromised by different types, shapes and





concentrations of soil microplastic (Lozano et al., 2021a). In addition, Boots et al. (2019) demonstrated reduced mean weight diameter of WSA in 30 days mesocosm experiments with plant and earthworm populations after application of juvenile mid-sized high-density polyethylene (HD-PE, 0.5-316 µm) and polylactic acid (PLA, 0.6-363 µm) particles as well as acrylic and nylon fibers. However, that data contrast with similar experimental set-ups, that showed no or even positive effects of juvenile fibers on WSA and WHC after up to 80 days of pot incubation (Lehmann et al., 2019; Zhang et al., 2019a; Lozano et al., 2021b; Qi et al., 2021). In addition, Liang et al. (2021) found WSA in a sandy loam soil unaffected by microplastic input, unless the test soil was amended with fresh plant material (0.8 wt%). The amendment caused increased aggregate formation, but also reduction of WSA by about a quarter compared to the control samples without microplastic.

**Table 1:** Effect of different microplastics on soil structural parameters. The abbreviations used in this table are as follows: frag – fragments, conc – concentration, incub – incubation time, POM – addition of particulate organic matter, %WSA – water stable aggregates, WSA – mean weight diameter of water stable aggregates, BD – bulk density, WHC – water holding capacity or field capacity, SHC – saturated hydraulic conductivity. Polymers: BP – bioplastic, PA – polyamide, PAA – polyacrylic acid, PE – polyethylene, PES – polyester, PET – polyethylene terephthalate, PC – polycarbonate, PS – polystyrole, PP – polypropylene. NA denotes that information was not available.

| soil texture | plastic type | plastic shape | particle size (µm) | pre-aging | conc. (mg kg⁻¹ dw) | incub. [d] | effect (mg kg⁻¹ dw) | reference |
|---|---|---|---|---|---|---|---|---|
| sandy clay loam | PE | NA | 102.6 (0.48−316) | no | 1000 | 30 | ↓ WSA diameter | Boots et al. (2019) |
|  | PLA | NA | 65.6 (0.6−363) | no | 1000 | 30 | ↓ WSA diameter |  |
|  | mixed | fibers | <2000 to >7000 | no | 10 | 30 | ↓ WSA diameter |  |
| loamy sand | PA | beads | 15−20 | no | 2500−20000 | ~35 | ↓ BD (>5000) | de Souza Machado et al. (2018) |
|  | PAA | fibers | 3756 (1260−9100) | no | 500−4000 | ~35 | ↓ BD (>500) |  |
|  |  |  |  |  |  |  | ↓ WHC (<1000) |  |
|  |  |  |  |  |  |  | ↓ %WSA (>0) |  |
|  | PE | frag. | 643 (160−1200) | no | 2500−20000 | ~35 | ↓ BD (>2500) |  |
|  |  |  |  |  |  |  | ↑ %WSA (>5000) |  |
|  | PES | fibers | 5000 (1540−6300) | no | 500−4000 | ~35 | ↓ BD (>500) |  |
|  |  |  |  |  |  |  | ↓ WHC (<1000) |  |
|  |  |  |  |  |  |  | ↑ WHC (>2000) |  |
|  |  |  |  |  |  |  | ↓ %WSA (>1000) |  |
| loamy sand | PA | beads | 15−20 | no | 20000 | ~70 | ↓ %WSA | de Souza Machado et al. (2019) |
|  | PE | frag. | 643 (mostly >800) | no | 20000 | ~70 | ↓ BD |  |
|  | PES | fibers | 5000 (1540−6300) | no | 2000 | ~70 | ↓ BD |  |
|  |  |  |  |  |  |  | ↓ %WSA |  |
|  | PET | frag. | mostly 222−258 | no | 20000 | ~70 | ↓ BD |  |
|  | PS | frag. | mostly 547−555 | no | 20000 | ~70 | ↓ BD |  |
|  | PP | frag. | mostly 647−754 | no | 20000 | ~70 | ↓ BD |  |
| loamy sand | PES | fibers | ~5000 | no | 1000 | 63 | no effect | Lehmann et al. (2019) |
| sandy loam | PAA | fibers | 370−3140 | no | 4000 | 42 | ↓ %WSA | Liang et al. (2019) |
| sandy loam | PA | fibers | ~5000 x 26 | no | 3000 | 42 | ↓ %WSA (POM) | Liang et al. (2021) |
|  | PES | fibers | ~5000 x 30 | no | 3000 | 42 | ↓ %WSA (POM) |  |
|  | PES | fibers | ~5000 x 8 | no | 3000 | 42 | ↓ %WSA (POM) |  |
| sandy loam | PA | fibers | <5000 | no | 1000−4000 | 42 | ↓ %WSA | Lozano et al. (2021a) |
|  | PC | frag. | <5000 | no | 1000−4000 | 42 | ↓ %WSA |  |
|  | PE | films | <5000 | no | 1000−4000 | 42 | ↓ %WSA (>2000) |  |
|  | PE | foams | <5000 | no | 1000−4000 | 42 | ↓ %WSA (1000 to <4000) |  |
|  | PES | fibers | <5000 | no | 1000−4000 | 42 | ↓ %WSA |  |
|  | PET | frag. | <5000 | no | 1000−4000 | 42 | ↓ %WSA (1000 to <4000) |  |
|  | PET | films | <5000 | no | 1000−4000 | 42 | ↓ %WSA |  |





| soil type | polymer | shape | size | | conc. | no. | effect | reference |
|---|---|---|---|---|---|---|---|---|
| | PP | fibers | <5000 | no | 1000−4000 | 42 | ↓ %WSA (all except 3000) | |
| | PP | films | <5000 | no | 1000−4000 | 42 | ↓ %WSA (<2000, >3000) | |
| | PP | frag. | <5000 | no | 1000−4000 | 42 | ↓ %WSA (1000 to <4000) | |
| | PS | foams | <5000 | no | 1000−4000 | 42 | ↓ %WSA (2000 and 4000) | |
| | PU | foams | <5000 | no | 1000−4000 | 42 | ↓ %WSA (3000) | |
| sandy loam | PES | fibers | ~1280 x 30 | no | 4000 | ~80 | ↑ %WSA | Lozano et al. (2021b) |
| silty sand | BP | films | 5000x5000 | no | 5000−20000 | 45 | ↓ BD (>5000) | Qi et al. (2021) |
| | | | | | | | ↑ WHC (>5000) | |
| | | | | | | | ↑ SHC (>5000) | |
| | | frag. | mostly 250−500 | no | 5000−20000 | 46 | ↑ WHC (>5000) | |
| | | | | | | | ↑ SHC (>10000) | |
| | PE | films | 5000x5000 | no | 5000−20000 | 43 | ↓ BD (>5000) | |
| | | | | | | | ↓ WHC (>10000) | |
| | | | | | | | ↑ SHC (10000) | |
| | | frag. | mostly 250−500 | no | 5000−20000 | 44 | ↓ BD (>5000) | |
| | | | | | | | ↓ WHC (>10000) | |
| | | | | | | | ↑ SHC (>5000) | |
| clayey loam (pot) | PES | fibers | NA | no | 1000−3000 | 47 | ↑ WSA diameter | Zhang et al. (2019a) |
| | | | | | | | ↑ pore space >30 µm | |
| | | | | | | | ↓ pore space <30 µm | |
| clayey loam (field) | PES | fibers | NA | no | 1000−3000 | 48 | ↓ WSA diameter (slightly) | |
| | | | | | | | ↑ pore space >30 µm (>1000) | |
| | | | | | | | ↓ pore space <30 µm (>1000) | |

In conclusion, the majority of data shows a negative effect of soil microplastic on WSA and
lead to the assumption that severe contamination of soils with microplastic can cause a loss
of soil structure and enhance erodibility. Some exceptions from these observations can be
explained by the respective soil environment. The increase of WSA mean weight diameter
with application of microplastic as observed by Zhang et al. (2019a) is assumed to be caused
by the extremely high clay content of the test soil (40%), which leads to aggregate formation
dynamics without major interference by the polymer particles. In contrast, soils with very low
clay content (1%) have minimum aggregation dynamics and, thus, show no influence of
microplastic addition on the formation of WSA (Qi et al., 2021). When aggregate formation is
accelerated through the introduction of strong aggregation agents such as the amendment
with fresh organic matter or earthworms, addition of microplastic cause significantly reduced
formation of WSA (Boots et al., 2019; Liang et al., 2021).

In contrast to WSA, soil hydrological studies on water holding capacity (WHC), saturated
hydraulic conductivity (SHC) and pore space distribution are sparse. Bulk density is overall
reduced by addition of the less dense polymers, but without clear relation to WSA nor WHC.
The higher WHC in some samples (de Souza Machado et al., 2018a; Qi et al., 2019) might be
caused by an increase of mesopore space (Zhan et al., 2019a), but clear statements cannot
be derived, yet, and more research on soil water balance characteristics after application of
microplastic is needed.

Overall data imply that microplastic type, shape and concentration as well as environmental
parameters such as vegetation, soil microbiome, and soil texture have influence on the
dimension of WSA loss (e.g. Lozano et al., 2021a; Boots et al., 2019; Zhang et al., 2019a;





Lehmann et al., 2019), but the underlying mechanisms are still not clear. The explanatory power of the data is restricted: All studies worked with very high microplastic concentrations of 500-20000 mg kg$^{-1}$ dry soil, which is 2 to 4 orders of magnitude above the concentrations in many soils and can only be found next to roads and on industrial sites (Dierkes et al., 2019;

Fuller and Gautam, 2016; Büks and Kaupenjohann, 2020). Furthermore, the studies exclusively used juvenile polymers, that have surface characteristics very different from weathered plastics. Aged microplastic in environmental samples may have different influence on soil aggregate formation and, thus, on parameters such as structural stability, carbon storage and water balance, which are strongly linked to soil fertility.

The above data, however, are helpful to hypothesize on the role of microplastic in aggregate formation. The different negative effects on soil structure can be explained by the fact that, in contrast to natural POM or clay minerals, the surface of unweathered microplastic is nearly uncharged (Table 2). Thus, the spatial integration of large amounts of microplastic fragments into soil structure lessens the cohesion of soil aggregates compared to POM particles at the

same place (Fig. 1). It can be hypothesized that this effect is enhanced with fibers, which provide a linear pattern of flaws, whereas small spheric particles have punctual influence. A laminar pattern as provided by films would even more act as a non-reactive barrier between natural soil particles with surface charge. This could explain, why larger and laminar particles are rather excluded from soil aggregates and have less influence on aggregation, while small

fragments and especially fibers were occluded (de Souza Machado et al., 2018). Even though most studies conducted short-term experiments within ≤48 days, prolonged incubation times could have lead to an initial alteration of juvenile plastic, a stronger integration into aggregates and can explain less decompromized soil structure in some experiments (de Souza Machado et al., 2019; Lehmann et al., 2019; Lozano et al., 2021b). Long-term incubations with lower

polymer concentrations, that allow to study changing surface characteristics of soil microplastic and the effect on soil structure under common environmental conditions, are still lacking to the best of our knowledge.



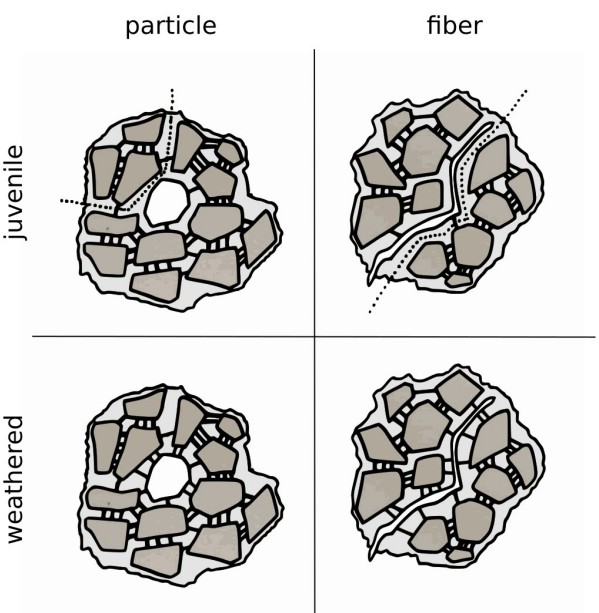

**Figure 1:** Simplified cross sections of a soil aggregate with occluded plastic particles and fibers in a juvenile and weathered state. Both mineral particles and particulate organic matter (POM) are represented by dark grey fragments, plastic items by white color. The number of interconnecting lines symbolizes the degree of physiochemical interaction, that keeps soil particles together. Dotted lines exemplarily stand for potential structural flaws resulting from a lack of bindings between juvenile plastic items and the surrounding soil matrix. The two cross sections below show a hypothesized stronger occlusion of weathered plastic particles.

## 4 Effects on soil biota

Plastic fragments in soils are also found to cause adverse effects on the edaphon and thereby affect soil functions related to soil structure, mass transport and metabolization. A review of 79 experiments, most of them conducted under laboratory conditions, was able to show that microplastic is ingested by the soil fauna in most cases (Büks et al., 2020a). The plastic further causes an alteration of the microbiome, digestive dysfunction, reduced body size and

reproduction, oxidative stress as well as inflammatory diseases in a variety of soil organisms. More recent experiments underline these observations (Kwak and An, 2021), but also show the contribution of extractable functional additives to the adverse effects (Kim et al., 2020). Although these results are restricted by the applied type, shape, degree of weathering, additives and concentrations of test microplastic, which often deviates from the characteristics





of aged environmental plastic fragments, small particle sizes are found to have by far the strongest effect on faunal health. Microplastic particles <100 µm, which provide a small fraction of mass, but high specific surface, caused adverse effects at concentrations of about
10 mg kg$^{-1}$ dry soil, whereas particle mixes with larger mean diameters mostly needed 1000 mg kg$^{-1}$ dry soil to attain similar effects (Büks et al., 2020b).

It is thereby possible, that even in trials with particle mixes of larger mean diameters the adverse effects are mainly caused by the small-sized fraction. In laboratory experiments, artificial microplastic is usually produced by extensive cryo-milling of polymer films (Büks et
al., 2020a). Due to hindrance of further mechanical comminution in this process, the <100 µm fraction appears to be <1 wt% beside coarser particles, as shown by Büks et al. (2021). It seems reasonable that laboratory experiments with only small-sized microplastic need lower mass concentrations to harm soil biota compared to experiments with microplastic of coarser diameter, since mainly the <100 µm fraction causes the adverse effect. However, in natural
soils items of <100 µm in diameter represent a larger mass fraction, that can amount to more than half of the soil microplastic in specific cases (Büks and Kaupenjohann, 2020).

Due to its smaller size, this fraction has a higher accessibilty to the gastro-intestinal tract of soil animals and may accumulate in inner organs. Its higher surface to volume ratio facilitates increased release of additives compared to a similar mass of larger items, which could partly
explain the negative effects of polymer particles in laboratory experiments with juvenile plastic (Büks et al., 2020a; Kim et al., 2020). In consequence, future experiments on the susceptibility of soil biota should focus on small-sized particles with aged surface characteristics to better distinguish between the genuine effect of environmental microplastic and secundary adverse effects caused by pristine concentrations of additives.

Beside the effects on soil structure and fauna, there are first indications that PE and PP microplastic affects soil metabolism by the alteration of microbial enzyme and respiration activities (e.g. Huang et al., 2019; Ng et al., 2020; Yi et al., 2020). These data are sparse and not yet attributed to specific surface characteristics.

Furthermore, the adsorption of persistent organic pollutants (POP) and heavy metals to soil
microplastic have been shown in several studies (e.g. Tourinho et al., 2019; Verla et al., 2019; Yu et al., 2020). However, there is an ongoing discussion, weather the sorption capacity of microplastic can contribute substantially to its adverse effects on soil organisms or have minor influence as a reservoir of toxic substances in face of the ubiquity and high amounts of natural POM and mineral surfaces in soils or even in aquatic systems (Koelmans et al., 2016; Verla et
al., 2019).





## 5 Quantification of microplastic surfaces

Together, impacts on soil aggregation, fauna, microbial biofilms and chemical adsorption
underline that the specific surface area is a promising candidate parameter for the prediction
of microplastic effects in soil. To date, the focus of microplastic quantification is on item
counting and masses (Bläsing and Amelung, 2018), while surface measurements were
applied by only a few authors. In different studies, BET analyses with $N_2$ were performed to
determine the specific surface area of aquatic microplastic samples and their alteration
through weathering (e.g. Wang and Wang, 2018; Zhang et al., 2018a; Fotopoulou and
Karapanagioti, 2012). Suchlike measurements in terrestrial environments are complicated as,
very similar to particle sizing, the quantification of microplastic surfaces requires a complete,
selective and non-destructive separation from the soil mineral matrix and elimination of POM,
that combines oxidative pre-treatment, mechanical agitation, density fractionation, oxidative
post-treatment and a method of surface determination (Kaiser and Berhe, 2014; Büks and
Kaupenjohann, 2020; Büks et al., 2021). This might be a reason, that BET measurements as
well as porosimetric methods were not yet adapted and applied to this task.

Existing quantifications with other techniques showed a surface area of 12.6±52.8 mm$^2$ kg$^{-1}$
dry soil (Zhang et al., 2020). In contrast, macrofragments >5 mm, that were picked from soil
samples, were found to additionally provide a much higher surface up to 21900±16700
mm$^2$ kg$^{-1}$ dry soil, which indicates an enormous potential of future soil microplastic supply
(Ramos et al., 2015; Zhang et al., 2020). However, these works with camera-microscopic
identification and counting are highly limited due to visual detection and the general exclusion
of fragments <50 μm. This implies an unknown underestimation of fractions with small particle
size and, thus, a large specific surface area. For that reason, an optimized design of surface
analyses with preceding complete and selective separation of microplastic from soil matrices
is crucial for any kind of analyses of soil microplastic surface characteristics. To date, the
development of such a procedure seems possible, since metal filters with a mesh aperture of
5 μm and strategies for selective sample preparation become available (Büks and
240  Kaupenjohann, 2020).

## 6 Weathering of plastic in the environment: The photooxidative and the biogeochemical phase

Provided that we do not only want to analyze the environmental soil microplastic, but also
245  simulate processes in the laboratory, experiments require the use of artificial plastic samples.
Hence, we have to take into account the alteration of microplastic surface characteristics due
to weathering processes. Most studies on the impact of microplastic on soil structure and
fauna used juvenile plastic, that has strong hydrophobic and uniform surfaces unlike material
that received natural or artificial aging (e.g. de Souza Machado et al., 2018; de Souza





250 Machado et al., 2019; Lehmann et al., 2019; Liang et al., 2019; Zhang et al., 2019a; Büks et al., 2020a). How does plastic aging in soils actually look like?

In a microscopic perspective, the surfaces of juvenile plastic items are normally smooth and uniformly structured with nearly no surface charge (Fotopoulou and Karapanagioti, 2012; Fotopoulou and Karapanagioti, 2015). When exposed to sunlight, which is mainly the case in 255 the "use and dispose" phase of the product life cycle, the weathering of plastic is largely driven by photooxidation. The incoming solar photons need to hit flaws (chromophores) within the polymer structure with wavelengths in the UV and blue spectrum to initiate photooxidative decay (Pickett, 2018). These reactions on impurities or structural groups like -NH- or aromatic rings along the polymer chain generate radicals, which cause chain scissions and reactions 260 with nearby polymers and $O_2$ resulting in crosslinks and a wide spectrum of carbonyl groups that increase surface charge (ter Halle et al., 2017; Dong et al., 2020). From the point of view of the macroscopic observer, the plastic becomes less hydrophobic, stiff, brittle and more prone to comminution. Further additives such as inks, plasticizers, flame retardants, UV absorbers and HALS (hindered amine light stabilizers) are degraded in parts also by longer 265 wavelengths of the UV-vis spectrum. The underlying reaction rates, except for the initial radical formation, increase with temperature and are also accelerated with advancing decay of chemical UV protection. This phase of weathering is well researched and reviewed (e.g. Kokott, 1989; Pickett, 2018), but it is not the final chapter.

When plastic is then exposed to the soil, the composition of weathering parameters changes 270 significantly (Table 2). The plastic is now faced to new mechanical stresses such as (bio)turbation and largely moist conditions that provide for biogeochemical attacks. One of these factors is the diverse and active soil fauna, that has been shown to ingest, digest and excrete plastic particles that fit to their gastrointestinal tract (Büks et al., 2020). Some taxa like woodlice, termites, mealworms and earthworms were additionally found to comminute plastic 275 by gnawing and, hence, actively produce microplastic (e.g. Lenz et al., 2012; Zhang et al., 2018b; Büks et al., 2020). In winter, when the mechanical treatment by biota is reduced, freeze-thaw-cycles might be an additional factor of comminution. Studies on the effect of alternating freezing and thawing on the structure of plastic surfaces are sparse and only focus on composite materials that include non-plastic components (Wang et al., 2007; Adhikary et 280 al., 2009; Zhou et al., 2014). However, water, that has already entered the cracks of weathered plastic with reduced hydrophobicity, most likely contributes to the comminution of the brittle material by freezing and expansion.

While moisture evaporates quickly on sun-exposed, heated plastic surfaces and is then not an important factor of weathering (Pickett, 2018), in soils it is the ubiquitous condition for 285 microbial life, extracellular metabolic processes and the release of chemical agents that react with the plastic. The microbial colonization and biofilm formation on surfaces of microplastic particles has been shown in studies on various aquatic ecosystems (e.g. Zettler et al., 2013; McCormick et al., 2014; Oberbeckmann et al., 2015; Dussud et al., 2018; Jiang et al., 2018).





Much scarcer in number, recent studies on soil ecosystems found surfaces of differently
originated microplastics inhabited by soil microbial communities, whose composition differs
widely from the soil matrix (Chai et al., 2020; Zhang et al., 2019b). This leads to a soil
microbial community altered due to microplastics application (Ng et al., 2020; Wang et al.,
2020). The population is thereby not only determined by the physiochemical properties of the
surrounding soil, but also by the type of plastic (Chai et al., 2020; Wiedner and Polifka, 2020;
Yi et al., 2020 ). In contrast, Yan et al. (2020) showed, that community composition as well as
P, $NO_3^-$ and $NH_4^+$ status of soils could be rather influenced by additives than polymer.

The degree of weathering also might be a control factor for biofilm cover, but to the best of our
knowledge, there are no studies on how biofilm development on plastic is affected by the
alteration of specific surface characteristics. Biofilm attachment as well as enzymatic
degradation are, however, supposed to be hindered by high hydrophobicity, low specific
surface area and smooth surface topography of plastic particles. Thus, advancing
photooxidative alteration of surfaces and brittle fracture might increase the formation of a
mature biofilm and the degradation of the plastic (Wei and Zimmermann, 2017).

Yet, we know that most bacteria have a net negative zeta potential (Tuson and Weibel, 2013).
Fotopoulou and Karapanagioti (2012) estimated the point of zero charge of beached
microplastic to be at pH 6.1, causing a negative charge in environments with higher pH values
due to the deprotonation of functional groups. Similar, plastic in soils with lower pH has a
positive surface charge and, thus, promotes the initial electrostatic attachment of cells. This
adhesive effect, however, is lessened in fluids with increasing ionic strength such as in soil
solution (Tuson and Weibel, 2013). Furthermore, both juvenile and weathered plastics adsorb
polar, non-polar and amphiphile organic molecules, some of them produced by the microbial
community, and develop a preconditioning film, that also affects surface charge. Also fungi
can use secreted hydrophobins to alter the hydrophobicity of a surface without its chemical
alteration and adsorb even on hydrophobic particles (Wessels, 1996). In consequence, the
type of outward-looking functional groups of the preconditioning film depends on the adsorbed
molecules and might "overwrite" the physiochemical properties of the plastic surface with
progressing development. The same cloaking mechanism can be expected during the
development of a mature biofilm. For this reason, it seems more difficult than expected to
estimate surface properties of colonized plastic particles from the degree of weathering.

A biofilm, in turn, causes the alteration of the plastic surface. Not only a viscous matrix, that
protects bacteria against mechanical stress, predators, desiccation and irradiation, it is also
an extracellular reaction space that facilitates the concentration and metabolization of
nutrients and the recycling of dead cell material (Flemming and Wingender, 2010). For this
purpose, manifold extracellular enzymes are produced by the biofilm community to
decompose food sources or modify the biofilm matrix in face of e.g. oxygen or nutrient
gradients (Flemming and Wingender, 2010). Among these are esterases, proteases and
amidases that target on substrates like polysaccharides, proteins, extracellular DNA, lipids





and urea, but also allow cometabolism of artificial polymers such as diverse polyesters, ester-based PU and PET (Shimao, 2001; Wei and Zimmermann, 2017; Danso et al., 2019). Yoon et
al. (2012) showed an unexpected degradation of PE by a bacterial alkan hydroxylase, and, beyond this, Yoshida et al. (2016) found the specific targeting of PET with a bacterial PETase. Polymers that have C-C backbones and no hydrolysable functional groups such as juvenile PE, PP, PS and PVC are assumed to be very slowly to hardly biodegradable by this groups of enzymes even in harsh environments. In contrast, unspecific lignin-degrading enzymes such
as laccases, manganese peroxidases, hydroquinone peroxidases and lignin peroxidases produced by actinomycetes, other bacteria as well as fungi, were shown to depolymerize even plastics such as PE, PS and PA, that were considered recalcitrant (Bhardwaj et al., 2013; Wei and Zimmermann, 2017).

In most of these cases, the observed decay gives no full evidence for polymer degradation. It
cannot be excluded that the measured weight loss during decomposition is caused by the degradation of additives, because many studies worked with commercial polymers, that have concealed compositions (Danso et al., 2019). However, beside the direct proof of enzymatic degradation pathways there are numerous references on the metabolization of (bio-)plastic samples by bacterial and fungal strains (e.g. Bhardwaj et al., 2013; Kale et al., 2015;
Raziyafathima et al., 2016; Roohi et al., 2017). In contrast, for PP and PVC neither degrading enzymes nor observed decay were reported (Danso et al., 2019). The practical side is, that enzymes, that have not been shown to target on plastics, are applied to purify extracted soil microplastic by degrading biofilms, that stabilize soil structure, or co-extracted organic matter (Büks and Kaupenjohann, 2016; Löder et al., 2017).

The microbial decay of microplastic does not only take place at plastic-biofilm interfaces within the soil pore space, but also within the soil fauna. Equipped with a diversity of masticatory organs, the edaphon does not only take part in the comminution of plastic objects as shown for woodlice, termites, meal-worms and earthworms (Büks et al., 2020a). It is also a multitude of small, mobile bioreactors, that incubate soil particles including microplastic within
a habitat of high microbial diversity – their gastrointestinal tract – and distribute them throughout the soil by excretion. A well known example for this multifaced functionality is the earthworm. There are also indications that the mealworm microbiome is able to degrade PE and PS to an eminent degree beyond the proportion of additives, but with yet no information on the underlying reactions (e.g. Brandon et al., 2018). In contrast, one-time short-term
exposition to gastro-intestinal enzymes might not be sufficient for such results. A sequential treatment of juvenile PE, PP, PVC, PET and PS to artificial human mouth, stomach and intestine exudates with amylase, protease and lipase for in total 155 minutes showed no significant alteration of size and shape of the microplastic particles, whereas the chemical alteration of surface properties was not measured (Stock et al., 2020). The gastro-intestinal
passage, however, is a complex mixture of degrading factors and is run through several times when plastic has entered the soil ecosystem.



Beside the soil biome, soil pH and oxidants are expected to directly influence the belowground alteration of plastic surfaces. While there is – to the best of our knowledge – no systematic examination of the effect of soil born acids, bases or oxidizing agents within
natural ranges of concentration and time of exposure, the treatment of plastic fragments with concentrated reagents caused damaging effects from color leaching and expansion to total dissolution (Enders et al., 2017). However, pre- and post-treatment with oxidants such as $H_2O_2$ are common parts of the extraction of microplastic from soil samples with density fractionation (Büks and Kaupenjohann, 2020). The agent is thereby used to degrade organic
matter that stabilizes soil aggregates in advance to the extraction of occluded microplastic or eliminates co-extracted particulate organic matter (POM) after the separation. A negative effect on plastic surfaces has not been ruled out in numerous applying studies.

**Table 2:** Development of surface characteristics during the three phases of aging (juvenile, photooxidative and biogeochemical phase). Data of biogeochemical weathering are only known from aquatic systems. (?) marks assumptions based on biogeochemical processes found in soils. Some references are: [1]Fotopoulou and Karapanagioti (2012), [2]Fotopoulou and Karapanagioti (2015), [3]ter Halle et al. (2017), [4]Dong et al. (2020), [5]Pickett (2018), [6]Andrady et al. (1993).

| characteristic | juvenile phase | photooxidative phase | biogeochemical phase |
|---|---|---|---|
| topography | smooth[1,2,4] | rough[5] | rough[1,2,4] |
| surface charge, carbonyl index | no[1,2,3,4] | yes[6] | increasing[1,2,3,4,(?)] |
| crystallinity, crosslinks, chain scissions | low[3] | high[5] | increasing[3,4,(?)] |
| biofilm cover | low | low | growing or mature[2,5,(?)] |
| aging factors | no | UV radiation[5] blue/violet spectrum[5] frequent leaching[5] | enzymes[(?)] organic acids[(?)] inorganic acids[(?)] bases[(?)] oxidants[(?)] bioturbation[(?)] feeding by the edaphon[(?)] frequent leaching[(?)] freeze-thaw-cycles[(?)] |

In conclusion, soil provides a variety of biogeochemical factors that cause continuous weathering of plastics. The majority of soil microplastic is therefore assumed to be biogeochemically weathered to a certain degree. The rate of this process, however, and its extent are still unknown, and some of the above fast alteration processes contrast with observations of very slow decomposition in soils (Bläsing and Amelung, 2018).





As soil environmental microplastic is actually altered, laboratory experiments require the application of plastic particles with similar surface characteristics. The particles have to be produced instead of collected, since the extraction from natural soils is unsuitable in many cases to provide the required microplastic, e.g. if pure plastic types, a defined degree of weathering or large amounts are needed. A number of studies in the last decades showed
significant differences between weathered and juvenile plastics and also between plastics that have been subject to photoxidative weathering under natural and artificial conditions with certain sources of radiation (e.g. Howard and Gilroy, 1969; Real et al., 2005; Friedrich, 2018; Dong et al., 2020). To the best of our knowledge, there is a lack of studies that compare the results of photooxidative close to nature techniques with those of belowground weathering.
Do similar characteristics arise from these two types of aging, or do we have to speak of three stages of weathering, the juvenile, the photooxidative and biogeochemical phase, that have to be taken into account in future soil microplastic experiments? And do current techniques of artificial weathering have the potential to alter microplastic similar to soil conditions?

**7 Artificial weathering for laboratory and field experiments**

A large group of treatments used for accelerated weathering of plastic surfaces originates from early materials science and industrial processes and includes an imitation of solar radiation by an UV or full-spectrum lamp, controlled temperatures and artificial irrigation with at least one of these factors enhanced compared to natural conditions (Pickett, 2018).
Treatments of several weeks cause severe weathering leading to enhanced crystallinity, density and cracked surfaces (Gulmine et al., 2003). Whereas formerly used carbon arc lamps are outdated because they emit a spectrum unlike natural sunlight (Howard and Gilroy, 1969), many industrial weathering protocols advice xenon arc lamps with borosilicate filters, that adjust the emitted spectrum tighter to the natural UV spectrum (DIN EN ISO 4892-2), or
fluorescent UV lamps (DIN EN ISO 4892-3). The performance of these lamps can be enhanced by use of modern daylight filters, a steady temperature of 38°C, relative air humidity of 25 to 50 % and regular washing of the sample surfaces by deionized artificial rain (Pickett, 2018). The equivalent incubation time corresponding to a certain period of natural weathering can be roughly estimated following Pickett (2018), but strongly depends on the
type of plastic. Standardized methods following DIN EN ISO 4892-2/3 are designed for studies on materials exposed to sun and weather, but are also currently applied in soil science approaches (BMBF initiative "Plastik in der Umwelt", e.g. Büks et al., 2021).

Beside the use of UV, the gamma irradiation is reported to imitate the carbonyl stretch in PE samples similar to a long-term exposition to UV-B radiation (Johansen et al., 2019).
Furthermore, Zhou et al. (2020) could demonstrate that discharged plasma oxidation (DPO) is likewise suitable to increase surface area, crystallinity and carbonyl indices of plastic particles within hours. However, plastic buried in soils is exposed to biogeochemical factors of



weathering different from the initial superficial exposition when entering the dimmed world of soil fauna, microorganisms, enzymes, organic acids, root exudates and frequent leaching.

The integration of biogeochemical factors into pre-weathering of artifical microplastic is considered only in a few studies (Table 3), alas fragmentary, heterogeneous and often directly applied to juvenile plastic. In experiments with soil organisms only a few authors pre-weathered the applied microplastic (Büks et al., 2020a). Tsunoda et al. (2010) heated plastic items within a water bath at 90 °C for 3 weeks and abraded the surface prior to feeding
experiments with termites. This treatment was aimed to make the surface more accessible for gnawing and might also extract soluble additives from the juvenile plastic. In another experiment, the formation of biofilms on microplastic surfaces was provoked by four weeks of incubation in seawater to make the material more attractive as a food source for the lugworm *Arenicola marina* (Gebhardt and Forster, 2018), an approach that can be likewise applied with
soil solution. With the intention to clean up artificial microplastic from soluble substances and fine particles, juvenile plastics were also treated with organic solvents such as methanol (Wang et al., 2019), ethanol (Rodrigues-Seijo et al. 2018; Rodrigues-Seijo et al., 2019) or pentane plus octane (Huerta Lwanga et al., 2016; Huerta Lwanga et al., 2017b; Yang et al., 2019). If the plastic type is prone to the solvents, the surface is roughened by the dissolution
of oligomers and, thus, increased. However, these techniques are not assumed to increase carbonyl groups and surface charge. Thus, they do not change the interaction with the soil matrix and the soil fauna, and were never tested on the similarity with natural weathering.

**Table 3:** Approaches of surface (pre-)weathering in recent experiments with soil microplastic. The abbreviations used in this table are as follows: UV – ultraviolet, TBBPA – tetrabromibisphenoal A, FE – feeding experiment. Polymers: BD – biodegradable plastics, OP – oxodegradable plastics, PA – polyamide, PE – polyethylene, PO – polyolefins, PP – polypropylene, PVC – polyvenyl chloride, TCE – thermoplastic copolyester elastomers.  NA denotes that information was not available.

| aging factor | applied plastic type | aging time (d) | resulting characteristics | experimental focus | reference |
|---|---|---|---|---|---|
| UV radiation (climate chamber) | diverse | variable | photooxidative aging | diverse | DIN EN ISO 4892-2, DIN EN ISO 4892-3 |
| gamma irradiation ($^{60}$Co source) | PE, PP | NA | photooxidative aging | cation adsoprtion | Johansen et al. (2019) |
| discharged plasma oxidation (DPO) | PVC | 0.02 | photooxidative aging | TBBPA adsorption of and toxicity to algae | Zhou et al. (2020) |
| wather bath (90°C) + abrasion | PO, PA, PE, TCE | 21 | extraction of additives, increased accessibility for feeding organisms | feeding experiment with termites | Tsunoda et al. (2010) |
| incubation in seawater | PA, PS | 28 | surface biofilm formation | FE lugworms | Gebhardt and Forster (2018) |
| incubation in aquatic systems | PE, PP | 19 | surface biofilm formation | cation adsorption | Johansen et al. (2019) |
| methanol treatment | PE, PS | NA | extract soluble additives | FE earthworms | Wang et al. (2019) |
| ethanol treatment | PE | NA | extract soluble additives | FE earthworms | Rodrigues-Seijo et al. (2018) |
|  | PE | NA | extract soluble additives | FE earthworms | Rodrigues-Seijo et al. (2019) |
| pentane + octane treatment | PE | NA | extract soluble additives | FE earthworms | Huerta Lwanga et al. (2016) |
|  |  | NA | extract soluble additives | FE earthworms | Huerta Lwanga et al. (2017b) |
|  |  | NA | extract soluble additives | FE earthworms | Yang et al. (2019) |
| plastic nursing (soil) | BD, OD, PE | ~150 | belowground weathering | mulch foil degradation experiment | Martin-Closas et al. (2016) |



| plastic nursing (soil, compost) | BD, PE | 14-365 | belowground weathering | feeding experiment with earthworms | Zhang et al. (2018b) |
| --- | --- | --- | --- | --- | --- |

Some authors avoided artificial weathering and instead applied natural aging over shorter periods of time, which can be used as a kind of "plastic nursing". Mulching films were aged between two weeks and 12 month by regular exposure in horticultures or buried into soils or composts (e.g. Martin-Closas et al., 2016; Zhang et al., 2018b). This treatment changes the physiochemical characteristics of plastics similar to environmental short-term weathering belowground and is suitable for aging large amounts of plastic, but might be very costly in terms of time when the production of strongly weathered microplastic is needed.

In conclusion, most studies either used a pre-weathering approach originated from materials science that only allows for aboveground alteration, or single surface editings on juvenile plastic that are aimed to simulate leaching, roughening and superficial biofilm formation, but still lack systematic justification. To the best of our knowledge, a full chain of aging – leaching of additives, photooxidative and biogeochemical aging – was never designed, tested or applied. The quality of future experiments with artificial microplastic will benefit from a standardized protocol that reproduces all stages of natural weathering.

## 8 Perspectives for future experiments

For decades, global soils received microplastic that is dispersed and occluded into the soil structure by biological and physiochemical processes. The particles underlie mechanical and biogeochemical alteration leading to a continual comminution and aging. This creates a growing fraction of weathered small-sized microplastic with low mass concentration (mg kg$^{-1}$ dry soil), but high number of items and large surface area (mm$^2$ kg$^{-1}$ dry soil), which is assumed to cause adverse effects on soil faunal health and influences soil structure more than larger particles. However, most studies on soil structure and soil biota – important attributes of soil health – worked with juvenile polymers, that have surface characteristics very different from aged plastic. Future research projects should therefore direct their attention to the measurement and reproduction of environmental soil microplastic surface characteristics in field and laboratory experiments. Based on the broad variety of degrading agents in soils and the fast formation of preconditioning films and biofilm cover on microplastic surfaces, the artificial alteration of soil microplastic surfaces is not sufficiently imitated by UV weathering, but requires an imitation of aging processes in soil. A standardized method of aging, that reproduces all phases of environmental weathering, will help us to precisely characterize the actual effects on microplastic on soil ecosystems. This method should include the photooxidative aging within climate chambers or nature, leaching of additives as well as biogeochemical weathering and surface biofilm formation.



**Data availability**

All of the data are published within this paper and in the Supplement.


**Author contributions**

FB developed the review concept, collected data and prepared the paper. MK supervised the study by participating in structural discussions on the idea and concept of the paper and the final corrections.


**Competing interests**

The authors declare that they have no conflict of interest.



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
