# Peer review of "The impact of microplastic weathering on interactions with the soil environment: a review."

_SOIL, 2021_

## Author Response (AR1)

Dear Referees.

Thank you very much for your mindful reviews and your very helpful comments. It has helped us to see some points which still need clarification. In the following, we want to explain how 5 we propose to adjust our article based on the comments and also explain, why in some cases we do not agree with the proposed changes.

First – and most important – we deeply agree with the referees, that the manuscript contains different topics, that are

- already known (introduction, photochemical weathering of plastic)

10 - already reviewed (influence of MP on the soil fauna)

- or not widely researched (MP and soil structure, biogeochemical weathering)

and that the focus of this work should be on the latter, but by use of a shorter, more perspective-based manuscript type.

Our response is split into two parts: This first part contains a restructured manuscript in the 15 format of a forum article with focus on soil biochemical weathering of microplastic surfaces and the possible need for soil-like pre-weathering of experimental microplastic. The second part (influence of microplastic on soil structure) will be re-submitted separately and is not part of this answer.

20 In the following you can find a list of all your points addressed (with numbers representing the order within the old document, excluded numbers no longer are part of the revised manuscript):

[5] Line 20-23: chemical reactions and physical processes are not clearly delineated; also, how do soil enzymes «weather» conventional plastics? The latter typically are chemically highly inert and it's not clear which enzymes 25 can act on these materials
→ We tried to clarify this within the abstract and discuss it later within the main text.

[7] Line 37: maybe fragmentation is the better term than comminution?
→ done across the manuscript
30
[9] Line 80: Why were studies on biodegradable plastics excluded? Why were papers on polymer photooxidation excluded (by requiring that the term "soil" was included in the search).

→ We focused on the named non-biodegradable polymers that represent …% of the plastic produced since the 1950s and their still growing legacy in soils. On the other hand, biodegradables are relatively new and only a 35 small part of the plastic introduced into soils.

[14] Line 140: The wording makes it sound as if the plastic is either "juvenile" or aged. However, the juvenile plastic will age when in soils. Also, it seems that the terms polymer and plastic are not clearly defined and used. They are not the same.

40 → That's correct. When juvenil plastic is added to the experimental soil ist starts aging. However, in short-term experiments it unlikely that even with the initial formation of biofilm cover there is an extended aging of surface characterists. To focus in the process, we replaced "aged" by "aging" in some cases. After shortening the text, "polymers" and "plastic, however, are now used in the correct way without any text modifications.

[24] Line 253: smoothness must depend on how the microplastic is formed/generated. Are the authors therefore sure that the microplastic is always "smooth"?

45 → We never had contrary impressions from REM/light microsopic images of juvenile commercial plastic items.

[25] Line 257: Chromophores are no "flaws". Also, there is indirect photolysis in which the polymers must not directly absorb light. Finally, most plastics that have exposure to sunlight are photostabilized. Photostabilizers slow down these reactions. This is not mentioned here.

→ Replaced by "weak bonds" and "indirect photolysis" added.

50 [26] Line 254: which of the conventional polymers contains NH groups?

→ deleted.

[27] Line 260: Carbonyls are uncharged

→ For soil environments, that's incorrect. Depending on the environmental pH, carbonyl groups (e.g. -COOH) are subject to (de-)protonation, which leads to variable charges. This is a process well known for soil organic
55 matter and the soil mineral matrix strongly controlling adsorption of molecules and interaction with other particles in soil.

[29] Line 270: what are "biogeochemical attacks"? And moisture is also present during the use period of the plastic.

→ Replaced: "The plastic is now faced to new mechanical stresses such as (bio)turbation, largely moist
60 conditions and exposed to a variety of biogeochemical processes."

[35] Line 339: Terms "decay" and "degradation" remain poorly defined. Aren't they describing the same overall loss of plastic integrity (either in terms of physical or chemical changes) / Line 384: the term "decomposition" is not defined. This is a general problem as the authors do not clearly define any of the terms. It seems that "weathering", decay, degradation, decomposition are all used interchangeably. Also, the term "biodegradation" is
65 not defined

→ Both, decay and degradation describe the breakdown of organic matter by the soil (micro-)biome, while in other parts of the text the word was replaced by "aging", "depletion" or otherwise clearyfied.

[32] Line 340: "weight loss" is misleading. Because there is also Mw (molecular weight). The reviewer assumes the authors refer to mass loss?

70 → Yes, thank you very much.

[37] Line 345: These polymers certainly decay. The authors mean that they don't biodegrade?

→ We instead used the term "biodegradation".

[38] Line 395: Why would one expect similar reactions? Photochemical reactions often trigger radical chemistry and needs light absorption and electron promotion to occur. This is not the case for subsurface reactions. So it
75 seems very unlikely that the very different reactions result in the same products (unless, of course, the chemistry is looked at in a blunt manner, eg: increase in "oxygen" content)

→ That's exactly our point: Pre-weathering of plastic for laboratory experiments is mainly conducted by use of climate chambers with photooxidation, but do not the outcome of aging in soil (that we do not know). We now have emphasized that point.

80   [39] Line 396: The reviewer cannot understand why "photochemistry" is separated from "geochemical" reactions. Aren't photochemical reactions also "geochemical"? For instance, according to Wikipedia (quick check, and not a scientific source, but most likely accurate here):·Photogeochemistry is the study of light-induced chemical reactions that occur or may occur among natural components of the Earth's surface.

→ For clarification we added "soil (bio)geochemical" throughout the document.

85   [40] Line 402: "early material science". What is meant by "early"? Photochemical aging of plastics is extremely well studied but does not seem to be "the early days" of material sciences

→ Deleted.

[41] Line 423: "dimmed world"? Why dimmed? Is this not "dark"?

→ Not necessarily. There can be faint light within the upper centimeters. And you can dim something until its
90 dark.

[42] Line 473: Is it reasonable to develop "THE" standard aging method for plastics in soils? See previous point

→ A standard approach, that includes influcenes by plastic type and additives but also so respective soil environment (e.g. arid/humid, active soil fauna, Corg).

[revised manuscript text omitted]

---

## Author Response (AR2)

Dear referees, thank you very much for your engagement in the re-review process and the nice and helpful advices. In the following you can find our modifications. Replies to the your comments are color marked with [numbers], further rephrasings (without changing the content) are only color marked.

Best regards,
Dr. Frederick Büks

**Answers to Referee #1**

[1] The statement that plastics age and therefore that fate and process studies in soils should ideally be done with aged plastics rather than juvenile (pristine/non-aged) plastics, seemingly the main point the paper argues for, seems obvious, raising the question if this point needs such a paper contribution.
→ There are a lot of studies that still use pristine plastic and neglect the influence of aged surfaces. However, this is not the focus of the present manuscript, but the additional contribution of underground weathering. The biogeochemical aging of MP surfaces in soil seems obvious due to the the transformative nature of soils, but there are some inconsistent indications that urge to study the extent of this effect: On the one hand, there is some work claiming extensive stability of plastic in soils and also works that show stability of MP in face of some stressors (e.g. Oberbeckmann and Labrenz, 2020; Büks et al., 2021), but on the other hand there are indications for susceptibilities discussed in this work. We hope to address this contradiction within the first part of chapter 3.

[2] While the manuscript addresses some of the possible weathering processes (in a rather rudimentary manner), statements or hypotheses for which processes in soils such aging may become relevant are not made. (...)
Line 59: this is a rather crude description of photochemistry. It's not that photons "hit weak bonds" thereby breaking them. In direct photolysis, photons are absorbed, an excited state is formed which can result in bond cleavage. In indirect photolysis, photochemically produced reactive intermediates form that can attack chemical bonds.
→ Since there is extensive literature on photooxidative weathering, the rudimentary description in our manuscript is now strongly condensed: "On a microscopic scale, the surfaces of pristine plastic items are normally smooth with nearly no surface charge (e.g. Fotopoulou and Karapanagioti, 2012; Fotopoulou and Karapanagioti, 2015). Exposed to sunlight, a depletion of UV absorbers and HALS (hindered amine light stabilizers) leads to enhanced photoxidation (e.g. Kokott, 1989; Pickett, 2018). From the point of view of the macroscopic observer, the plastic becomes less hydrophobic, stiff and more prone to fragmentation by wind and water erosion."
We further specified soil processes, that might be affected by changing surface properties of MP: soil structure, edaphon health, transport of plastic and soluble substances (see first paragraph of chapter 3).

[3] Line 18: Any plastic particle entering soils will undergo additional changes on its surface. It seems that this is not a mere possibility but, in fact, a given!

→ We specified, that we don't know how extensive these changes are: "When plastic particles then enter the soil environment, further aging factors appear with yet unknown efficacy."

[4] Line 19: Decay with enzymes seems limited to specific plastics whereas conventional plastics are resistant to enzymatic attack. This needs to be expressed more clearly considering recent studies claiming that conventional
plastics are enzymatically degradable. This is simply not the case.
→ Wei and Zimmermann (2017) and others reviewed experiments, that showed the degrading effect of enzymes on conventional and biodegradable plastics (see line 121-131). Only the magnitude of the effect in complex systems is unclear. We therefor added "(… with both conventional and biodegradable plastics), …" to line 19.
[5] Biotic and abiotic acids? What is meant here? Grammatically incorrect it seems.
→ Right you are. We changed the sentence: "…, contact with biotic and abiotic acids, oxidants as well as uptake by the soil fauna that causes physical fragmentation." Biotic acids include root exudates, abiotic acids are e.g. carbonic acids or nitric acid. In both cases, it
cannot be ruled out that they lead to long-term weathering of embrittled plastic (although we know, that e.g. laboratory equipment is very stable in face of most of the acids).

[6] Line 20: it clearly is desirable to work with plastic objects that mimic those in nature. But for which types of experiments is this relevant? For instance, persistence of the particles unlikely is changed by these
modifications. Transport characteristics also not for larger particles (but maybe for nano-sized particles). (…) The reviewer would have appreciated a bit more guidance as to which processes are affected rather than a relatively obvious statement that pristine plastic does not equal aged / weathered plastics.
→ We added "Such transformation of surfaces is assumed to affect soil aggregation processes, soil faunal health and the transport of plastic colloids and adsorbed solubles." to
line 21 (see also [2]).

[7] Is it possible that all plastic "looks alike" in soils if chemically different materials obtain the same "coating" (ecocorona)?
→ This is really a very interesting point. We included it by changing lines 113-119 as follows:
"Recent studies on soil ecosystems have also demonstrated that MP surfaces of different origin are covered with microbial communities. This might hypothetically cause a masking of plastic surface characteristics by the biofilm matrix. The composition of surface MP communities is very different from that of the soil matrix (Chai et al., 2020; Zhang et al., 2019). The altered soil microbial community is thereby not only determined by the
physiochemical properties of the surrounding soil, but also by the type of plastic and its additives (Chai et al., 2020; Ng et al., 2020; Wang et al., 2020; Wiedner and Polifka, 2020; Yan et al., 2020; Yi et al., 2020). This might lead to a physiochemical behavior of plastic particles, that differs not because of the plastic type, but because of its biofilm cover."

[8] Line 24: What is meant with "young"? Can plastic be young? The authors mean pristine? Non-aged?
→ Thank you. We now use "pristine" in the sense of "not aged" instead of other terms throughout the manuscript.

[9] Line 41: Sure. But these are studies that used plastics in the mass% range? How realistic is this? And is this
not a rather trivial finding that a soil with, lets say 10 mass % of fine ground plastic, is no longer behaving like a soil without 10% of plastic? A soil with 10% more sand or clay (or anything) will change its properties as well.

→ We strongly agree, that – especially in experiments with massive addition of MP – a clear distinction between adverse effects caused by the MP itself and those caused by changing physicochemical conditions is necessary. However, from our point of view, this is beyond this forum article.

[10] Line 51: it is true that plastics exposed to the elements undergoes weathering. However, many of the plastics contain additives to prevent chemical transformation. UV stabilizers, pigments, antioxidants … . Is it possible that for many plastics photochemical weathering is small because of these additives? Also, given that plastics differ not only in polymeric composition but also types and concentrations of protective additives, what do the authors suggest? Case-specific weathering of commercial items? Clearly, weathering PE from vendors like Sigma Aldrich or Fisher Scientific, etc would not help if these are, for instance, not photostabilized.
→ We added "depletion of UV absorbers and HALS (hindered amine light stabilizers)" to [2] to clarify, that aging is hindered as long there is intact protection.

[11] Line 54: what is a microscopic perspective?
→ Better: "On a microscopic scale …"

[12] Line 55: "Uniformly structured" seems a simplification: for instance, semicrystalline polymers contain amorphous regions and crystalline lamella which behave very differently.
→ You're right, AFM images clearly show this oversimplification. We deleted "... and uniformly structured …"

[13] Line 58: "photooxidation and indirect photolysis". This is incorrect. Photooxidation is a chemical reaction that can result from both direct and indirect photolysis.
→ See modification in [2].

[14] Line 79: recommend instead of advice
→ Thank you. Done.

[15] Line 83: is gamma irradiation a relevant aging process in the environment?
→ Fortunately not, but used to accelerate aging. To avoid misconceptions: "γ-irradiation treatment".

[16] Line 85: well, any oxidative process would, no? there are many ways of introducing surface oxygen functionality into polymers.
→ Sure, but this is the recent work on this topic.

[17] Line 93: why "dimmed". Not dark?
→ In the first mm of soil there are more dimmed than dark condition. But dark is largely correct. Done.

[18] Line 100: Why is soil fauna a "bioreactor"? Sure, they are "alive" and hence "bioreactors". But most plastics will not be affected chemically when passing through these organisms. If humans swallow stones, would it be correct to say they are "bioreactors"? The reviewer would argue that the stone comes out as it went in (except for some coating). Bioreactors seems to invoke the false impression that the plastic is significantly processed.
→ The intestinal tract of the soil fauna provides an environment with conditions enhancing the activity of the soil microbiome. Here, MP is covered with biofilms and occluded into casts. These fundamental processes justify this terminology, although the efficacy of plastic aging in the GIT is unknown.

[19] Line 105-106: Does this make PS and PE non-persistent? By no means.
→ If additives are depleted, the polymer is more susceptible to the environment. And if then the polymer is degraded, it is clearly a sign of non-persistence. See modification [33].

[20] Line 106: First, there is relative humidity: a chemical reaction involving water does not need "pure liquid". Second, which reaction can water perform on conventional plastics? Hydrolysis reactions? No.
→ Thank you. We deleted "… and is then not an important factor of weathering (Pickett, 2018), …" In the soil, liquid water plays a major role for nearly all aging processes in question.

[21] Line 114-115: Any surface in a soil will lead to enrichment of specific microorganisms. This per se is not a surprising finding. A leaf added to soil will have a surface microbial community that differs from that in the soil.
Sure. From our point of view, there is no reason to skip this point within the discussion of surface alteration and masking (see [7]).

[22] Line 134-137: But these polymers remain recalcitrant even if some enzymes (in lab incubations?) oxidize some parts of these polymers.
→ Oxidation of parts of the plastic surface causes surface alteration. If the oxidation is going on, there is an increasing alteration, and this leads to long-term degradation. But of cause, we have to mention the laboratory character of the experiments: "Given a poor biodegradability of polymers with C-C backbones and no hydrolysable functional groups such as juvenile PE, PP, PS and PVC, laboratory experiments showed an unexpected
degradation of PE by a bacterial alkan hydroxylase (Yoon et al., 2012), and, beyond this, the specific targeting of PET with a bacterial PETase (Yoshida et al., 2016)."

[23] Line 148: soil pH affects the surfaces? How so? Do they contain pH-sensitive acid/base groups?
→ Thank you, the sentence was mistakable. Not variable charge, but "soil born acids and oxidants".

**Answers to Referee #2**

[24] It still has lengthy paragraphs of review character that should be shortened to the information relevant to substantiate the authors' view point.
→ Thank you, we condensed the descriptions of photooxidative aging (see [2]) and artificial weathering (line 70-91).

[25] The main title and the section titles do not reflect the article's view point character and the authors' intent to call for a new methodologic approach in the microplastics research community.
Title should reflect the intention of the paper: an appeal to where research should be directed.
→ We changed the title to "What comes after the sun? – On the integration of soil biogeochemical pre-weathering into microplastic experiments". We also merged the two weathering chapters to "Underground weathering – a second phase of aging?" and renamed the conclusion to "Pre-weathering under soil conditions: A methodology for future
approaches?"

[26] Consider to use the more common term "pristine" instead of "juvenile"
→ Done (see [8]).

**[27]** Some expressions/wordings are a little clunky in parts and quite a few grammar errors and typos occur throughout. I'd recommend a revision by a native speaker to ensure the language is easy to read and inambiguous, and also to shorten the sentences.

→ Thank you very much for that advice.

**[28]** Line 13: replace "MP fraction" with "MP size fraction"

→ Done.

**[29]** Line 24: why this title? Rather, a point should be made that studies so far have widely ne-glected the dimension of time of plastic exposure to environmental factors.

→ We changed the section title to "Did we neglect biogeochemical aging factors?"

**[30]** Line 53 ff: instead of the following two sections divided into photooxidative and "biogeo-chemical" weathering, I would recommend to craft this into one section. In such, the authors can get to the point, using less words, which types of weathering relevant for soil environments have been considered in past studies and which have not.

→ Done (see [25]).

**[31]** That said, mechanical effects that have been assigned solely to the underground part (that is how I in-terpret "dimmed world of soil fauna…") might happen as well at the surface, e.g. during abrasion with soil particles caused by wind/water erosion.

→ See [2].

**[32]** Line 72: Please provide a link to this initiative and translate for the international reader-ship

→ Done: "… Plastic in the Environment", https://www.bmbf-plastik.de/en, ...).

**[33]** Line 105: I am not sure I understand the meaning of "to an eminent degree beyond the proportion of additives"

→ "There are also indications that the mealworm microbiome is able to degrade not only additives, but also PE and PS polymers."

**[34]** Lines 137 ff: I am missing the argument that plastic is unlikely to serve as a major sub-strate for microbes due to their large molecular size, high chemical stability, and low bio-availability (Oberbeckmann and Labrenz 2020)

→ We added "Although plastic is unlikely to serve as a major substrate for microbes due to its large molecular size, high chemical stability, and low bioavailability (Oberbeckmann and Labrenz 2020), there is indication that a biofilm causes the alteration of MP surfaces." to line 118.

**[35]** Table 1: Why put (?) if there are references?

→ Sorry, leftover. Deleted.

**[36]** Line 156: replace "mechanical treatment through biota" with "biotic effects".

→ Done.

**[37]** Line 163 ff: This title is not very meaningful, and also this section has too much "review-character". For a view point paper, the authors should focus more on what they think is needed for future studies, but not putting too much detail into the techniques of weathering simulation that have been applied in the past.

→ We added the focus on future practice (see [1]), but prefer to keep the description of past techniques, since there is no work that collects those information.

[38] 1. If this was the case, would it not have tremendous implications on the quantification methods currently used? The stresses are presumably much stronger in the extraction set-ting compared to a soil in situ. In fact, can some of the separation methods be considered accelerated weathering (e.g. breaking down soil aggregates in a mortar, using acidic rea-gents, enzymatic digestion)? Or can we assume that these effects are negligible in situ, if microplastic particles resist the harsher conditions in the lab?

→ That is a very interesting point. From our point of view, cavitational stress via ultrasound and e.g. gnawing are not the same type mechanical stress and, thus, stability of MP in face of ultrasound as shown in Büks et al. (2021) is not representative for feeding on MP. A discussion of that point, I think, would exceed the format of the forum article.

[39] 2. However, in the case that microplastics are altered to a significant degree once they enter below the soil surface, why should weathering simulation methods be established before we even know if these processes alter microplastics behavior and impacts, e.g. on organisms or soil structure? Should the community not prioritize on effect studies that compare pristine with weathered particles regarding their effects on soil biota, accumulation, transport, etc.? What about an alternative approach of studying the effects in soils that have been contaminated under real conditions in field studies of soils with high "legacy" microplastic contamination and controls with low levels (e.g., in long term sewage sludge or compost trials)?

→ Thank you. We tried to address this step-by-step approach throughout the manuscript.

[revised manuscript text omitted]

---

## Author Response (AR3)

Dear John Quinton, thank you very much for your suggestion of an alternative title that more closely reflects the focus on soil biogeochemical weathering. We prefer to keep the title "What comes after the sun? – On the integration of soil biogeochemical pre-weathering into microplastic
experiments", as - from our perspective - "What comes after the sun" sufficiently indicates that we are talking about a process that follows the aboveground UV weathering. We hope that you can agree to this.

Dear Peter Fiener, thank you very much for your dedicated assistance in this complex review process.

[revised manuscript text omitted]